# How alginate properties influence in situ internal gelation in crosslinked alginate microcapsules (CLAMs) formed by spray drying

Tina Jeoh[1☯*], Dana E. Wong[1☯¤a], Scott A. Strobel[1☯¤b], Kevin Hudnall[1‡], Nadia R. Pereira[2‡], Kyle A. Williams[3], Benjamin M. Arbaugh[1], Julia C. Cunniffe[1¤c], Herbert B. Scher[1‡]

1 Department of Biological and Agricultural Engineering, University of California, Davis, CA, United States of America, 2 Laboratory of Food Technology, Universidade Estadual do Norte Fluminense Darcy Ribeiro, Campos dos Goytacazes, RJ, Brazil, 3 Malvern Panalytical, Westborough, MA, United States of America

☯ These authors contributed equally to this work.
¤a Current address: DuPont Nutrition & Biosciences, Palo Alto, CA, United States of America
¤b Current address: PivotBio, Berkeley, CA, United States of America
¤c Current address: USDA-ARS, Western Regional Research Center, Albany, CA, United States of America
‡ These authors also contributed equally to this work
* tjeoh@ucdavis.edu

**Data Availability Statement:** All relevant data are within the manuscript and its Supporting Information files.

## Abstract

Alginates gel rapidly under ambient conditions and have widely documented potential to form protective matrices for sensitive bioactive cargo. Most commonly, alginate gelation occurs via calcium mediated electrostatic crosslinks between the linear polyuronic acid polymers. A recent breakthrough to form crosslinked alginate microcapsules (CLAMs) by in situ gelation during spray drying ("CLAMs process") has demonstrated applications in protection and controlled delivery of bioactives in food, cosmetics, and agriculture. The extent of crosslinking of alginates in CLAMs impacts the effectiveness of its barrier properties. For example, higher crosslinking extents can improve oxidative stability and limit diffusion of the encapsulated cargo. Crosslinking in CLAMs can be controlled by varying the calcium to alginate ratio; however, the choice of alginates used in the process also influences the ultimate extent of crosslinking. To understand how to select alginates to target crosslinking in CLAMs, we examined the roles of alginate molecular properties. A surprise finding was the formation of alginic acid gelling in the CLAMs that is a consequence of simultaneous and rapid pH reduction and moisture removal that occurs during spray drying. Thus, spray dried CLAMs gelation is due to calcium crosslinking and alginic acid formation, and unlike external gelation methods, is insensitive to the molecular composition of the alginates. The 'extent of gelation' of spray dried CLAMs is influenced by the molecular weights of the alginates at saturating calcium concentrations. Alginate viscosity correlates with molecular weight; thus, viscosity is a convenient criterion for selecting commercial alginates to target gelation extent in CLAMs.

**Funding:** The authors received no specific funding for this work. The USDA National Institute of Food and Agriculture's Agriculture and Food Research Initiative provided support in the form of salary for author SAS (grant 2018-67012-28029). Malvern Panalytical provided support in the form of salary for author KAW. Malvern Panalytical provided the infrastructure to collect GPC/SEC data for the samples in this study. The funders did not have any additional role in the study design, data analysis, decision to publish, or preparation of the manuscript. The specific roles of these authors are articulated in the 'author contributions' section.

**Competing interests:** TJ's research group has received research support from BASF. Malvern Panalytical's involvement in facilitating GPC/SEC data collection does not alter our adherence to PLOS ONE policies on sharing data and materials. KAW is an employee of Malvern Panalytical. TJ, SAS, and HBS are inventors on the 'CLAMs process' patent US9700519B2; TJ, BMA, and HBS are inventors on a pending patent applying CLAMs to encapsulate bacteria (US Patent Application 63,033,305); TJ, SAS, DW and HBS are inventors on a pending patent for a process improvement to the CLAMs process (US Patent Application 62/775,331); TJ, SAS and HBS are inventors on a pending patent applying the CLAMs process to encapsulate oxygen labile cargo (US Patent Application 62/837,399). The authors' inventor status does not alter our adherence to PLOS ONE policies on sharing data and materials.

## Introduction

Alginate is a linear polyuronic acid derived from macroalgae and bacteria with useful industrial applications due to its ability to form reversible crosslinked matrices in the presence of divalent cations [1–3]. Microencapsulation in crosslinked alginate provides long-term shelf stability and controlled delivery of bioactives in a broad array of industries. For example, encapsulation of pharmaceuticals in crosslinked alginate can facilitate drug delivery, wound healing and cell transplantation [2,4]. Crosslinked alginate can protect oxygen sensitive ingredients, mask unwanted flavors and confer enteric delivery of nutrients and nutraceuticals in foods [5–8]. Crosslinked alginate encapsulation can increase shelf stability and facilitate controlled release of Gram negative plant beneficial bacteria on seed or in soil for sustainable industrial agriculture [3,9]. The success of encapsulation in alginate in these and other applications relies on being able to control the matrix properties by regulating gelation during formation.

Typically mediated by calcium, gelation by electrostatic crosslinking of alginates is rapid at physiological conditions, and reversible. Crosslinking imparts insolubility to the alginate matrix in water, but the resulting gel can easily dissolve in the presence of chelators that sequester calcium ions. Crosslinked alginates can be selective barriers, controlling exiting rates of encapsulated cargo or entering rates of moisture or oxygen. One effective handle to control alginate matrix barrier properties in both the dry and hydrated state is by regulating the extent of crosslinking [9,10]. Higher crosslinking extents can decrease swelling/water uptake and slow matrix erosion in water, thus minimizing diffusion losses of the encapsulated cargo [10–12].

Ion-mediated crosslinking of alginates can be achieved by internal or external gelation [13], where crosslinking extents can be controlled at the process level by varying calcium salt concentrations and alginate-calcium contact times [14]. Commonly used gelation processes, however, require multiple steps, long processing times and specialized equipment [15–17]. Difficulties and cost of scaling the process limits the commercial use of crosslinked alginate encapsulation despite its wide research presence [18,19]. A recent breakthrough enabling in situ gelation during spray drying, however, paves the way for low-cost industrial-scale crosslinked alginate microcapsule (CLAMs) production [18,20]. The "CLAMs process" uses a calcium salt that is insoluble at the feed pH to prevent gelation before spray drying; when the finely divided suspension is spray atomized, a pH drop due to volatilization of the base in the formulation solubilizes the calcium salt and releases calcium ions to facilitate crosslinking of the alginates. Thus, the multi-step process of particle formation, crosslinking and drying is collapsed into a single, industrially ubiquitous spray drying operation [18]. The mechanisms of particle formation and drying in the spray dryer is well-documented and recently modeled specifically for the CLAMs process to show exponential moisture removal during drying [21]. Alginate gelation in CLAMs is controlled by the concentration of the calcium salt in the feed formulation [10].

In addition to process controls, alginate properties are well-documented to impact its gelling propensity [1,22–24]. While alginates are commercially available for purchase, the alginate product is typically sold with little to no details of the molecular properties. The information provided with commercial alginates is generally limited to its bulk viscosity in aqueous solutions and recommended use (e.g. Table 1). Commercial alginates are primarily extracted from macroalgae and processed to varying degrees [4,25], influencing the molecular size and composition of the alginate product [26,27]. The ambiguity of the properties of alginates complicate the predictability of resulting CLAMs matrix properties.

The aim of this work was to examine how molecular properties of alginates impact crosslinking in CLAMs formed by in situ gelation during spray drying. While molecular

composition impacts crosslinking in externally gelled alginates, we hypothesized that cross-linking in spray dried CLAMs may be limited by different factors. We characterized a selection of commercial alginates and examined the influence of molecular weight, size distribution and composition on the extent of crosslinking in CLAMs. The CLAMs were also examined by Fourier transformed infrared (FTIR) spectroscopy to confirm crosslinking. Ultimately, this work aimed to understand the appropriate criteria for selecting commercial alginates to influence the extent of crosslinking in CLAMs.

## Materials and methods

### Materials

Sodium alginate from Millipore Sigma (Cat# A1112) (LV), BASF (Hydagen 558P) (HV1), TIC (Algin 400) (HV2), and DuPont Danisco (GRINDSTED Alginate FD 155) (HV3) were used in this study (Table 1). Calcium hydrogen phosphate, succinic acid, sodium citrate, glacial acetic acid, sodium carbonate, β-D-glucose, sodium bicarbonate, L-serine, sodium hydroxide, ammonium hydroxide, hydrochloric acid, calcium carbonate, and methanol were purchased from Thermo Fisher. Schiff's fuchsin sulfite reagent, sodium metabisulfate, periodic acid, anthrone, D-(+)-galacturonic acid, concentrated sulfuric acid, disodium 2,2-bicinchoninate (BCA), copper (II) sulfate, chloroform, 1-phenyl-3-methyl-5-pyrazolone (PMP), and D-(+)-mannuronic acid were purchased from Millipore Sigma. The polymannuronic acid (YP31737) and polyguluronic acid (YP03135) was obtained from Carbosynth Ltd. (Berkshire UK) at a minimum of 85% purity and 10% water content with an average molecular weight of 6–8 kDa.

### Methods

**Characterization of alginates.** *Alginate molecular weights.* Commercial alginate samples were dissolved to 5 mg/mL in aqueous 0.05 M sodium sulfate at ambient temperature with gentle rocking overnight, and filtered through a 0.2 μm nylon syringe into autosampler vials. The molecular weight distributions of the alginate samples were determined by size exclusion chromatography with refractive index, in line viscometer, and right angle/low angle light scattering detectors (OMNISEC SEC/GPC Malvern Panalytical). Samples were separated in 0.05 M sodium sulfate as the mobile phase, using a set of 2 x A6000M 300x8mm columns, flow rate of 1.0 mL/min, 100 μL injection volume, column/detector temperature of 25°C, and autosampler temperature of 20°C.

*Guluronic acid to mannuronic acid (G/M) ratio in alginate samples.* Average G/M ratios for the alginates were determined from measured concentrations of guluronic (G) and mannuronic (M) acid residues in fully hydrolyzed samples. Alginate hydrolysis was conducted by two-step acid hydrolysis where 0.3–1.0 g of alginates were mixed with 3 mL of 72% sulfuric acid in glass pressure tubes in a 30°C water bath for 1 h, and stirred every 10 min. Subsequently, 84

**Table 1. Commercially sourced alginates used in this study.**

| Alginate ID | Commercial Source and Name | Manufacturer informed characteristics and/or use |
|---|---|---|
| **LV** | Sigma A1112 | Low viscosity |
| **HV1** | BASF Hydagen 558P | Thickening agent for cosmetic preparations |
| **HV2** | TIC-Algin ® 400 | Medium viscosity; gelling agent for food |
| **HV3** | DuPont Danisco GRINDSTED ® Alginate FD155 | Gelling agent for food |

mL of water was added to each tube before autoclaving at 121˚C for an hour. A sugar recovery standard containing predetermined concentrations of glucose and galactose was processed in parallel to account for degradation losses. After cooling, 20 mL of DI water was added to the hydrolysate, then vacuum filtered through Gooch crucibles which had been dried overnight at 575˚C. The hydrolysate was neutralized with calcium carbonate to pH 5–6, then filtered through 0.45 μm syringe filters and stored at 4˚C.

The uronic acids in the filtered hydrolysates were tagged with 1-phenyl-3-methyl-5-pyrazo-lone (PMP) to increase the sensitivity by UV detection [28,29]. A 500 μL sample of neutralized hydrolysate was mixed with 500 μL of 0.3 M sodium hydroxide and 600 μL of 0.5 M PMP (in methanol). After incubation for 1 h at 70˚C and cooling, samples were re-neutralized with 500 μL of 0.3 M hydrochloric acid. Tagged samples were separated and extracted with three rounds of 2.5 mL chloroform. The aqueous layer containing solubilized uronic acids was removed and filtered through 0.45 μm syringe filters before HPLC analysis.

Monomer concentrations in hydrolyzed alginate samples were analyzed by HPLC (Shimadzu Scientific Instruments, Columbia, MD) with a BioRad Aminex HPX-87P column (BioRad, Hercules, CA) and Carbo-P guard column deashing, at 0.6 mL/min pump rate, 80˚C and filtered nanopure water mobile phase. Refractive index and PDA(UV) detectors monitored the sample retention times compared to tagged mannuronate standards in DI water between 0 μg/mL to 1 μg/mL.

*Alginate solution density and viscosity measurements*. Alginate solution density was determined as the ratio of the mass of 50 mL of alginate solution in a 50 mL volumetric flask to 50 mL volume. Alginate solution viscosities were measured using a Gilmont, size No. 3 Falling-Ball Viscometer (Cole-Parmer, Barrington, Illinois) with a size 3 glass or stainless steel ball according to manufacturer specifications.

**Crosslinked alginate microcapsules (CLAMs) formation.** *CLAMs formed by internal gelation during spray drying.* Spray-dried CLAMs were produced as previously described [6,20]. Feed suspensions were prepared by mixing calcium hydrogen phosphate with succinic acid titrated to pH 5.6 or pH 8 with ammonium hydroxide, and alginates hydrated for $\geq 40$ min. Succinic acid was at half the concentration of alginate. CLAMs using HV1, HV2 or HV3 alginates were made with a feed solution of 0.5% (w/w), while CLAMs using LV alginates were at a feed solution of 2% (w/w). Feed suspensions contained mass ratios of 0.25, 0.125, 0.1, 0.083, or 0.05 calcium phosphate to alginates.

CLAMs were formed in a Büchi B-290 benchtop spray dryer (New Castle, DE) using an inlet temperature of 150˚C, aspirator air flow at 35 m$^3$/hr, feed peristaltic pump at 20% of the maximum, and air nozzle flow at 40 mm (Fig 1). Outlet temperatures were 85–95˚C under these conditions. The dry powder samples were stored in desiccators until analysis.

*Alginate crosslinking by external gelation.* External gelation of alginates was carried out by dripping a 1% solution of polymannuronates (PolyM), polyguluronates (PolyG) or varying ratios of PolyM and PolyG alginates at pH 5.8 into a solution of 0.25% calcium chloride in a beaker at room temperature [21]. Alginate solutions were dripped from a burette into a slow and continuously stirred calcium chloride solution on a stir plate. The rate of dripping was controlled to form distinct drops into the solution. The final contents in the beaker was targeted to 0.5% alginate and a calcium to alginate ratio of 0.25.

The entire solution was poured into petri dishes and dried at 50˚C for 48 hours in an oven. Dried solids were collected in 50 mL conical tubes with a small stir bar, and rotated for 2 h on a benchtop rotator to generate homogenous powders. The powders were stored in a desiccator at room temperature.

**Characterization of CLAMs.** *Extent of crosslinking–soluble alginate assay.* The extent of cross-linking in CLAMs and alginates crosslinked by external gelation was determined as

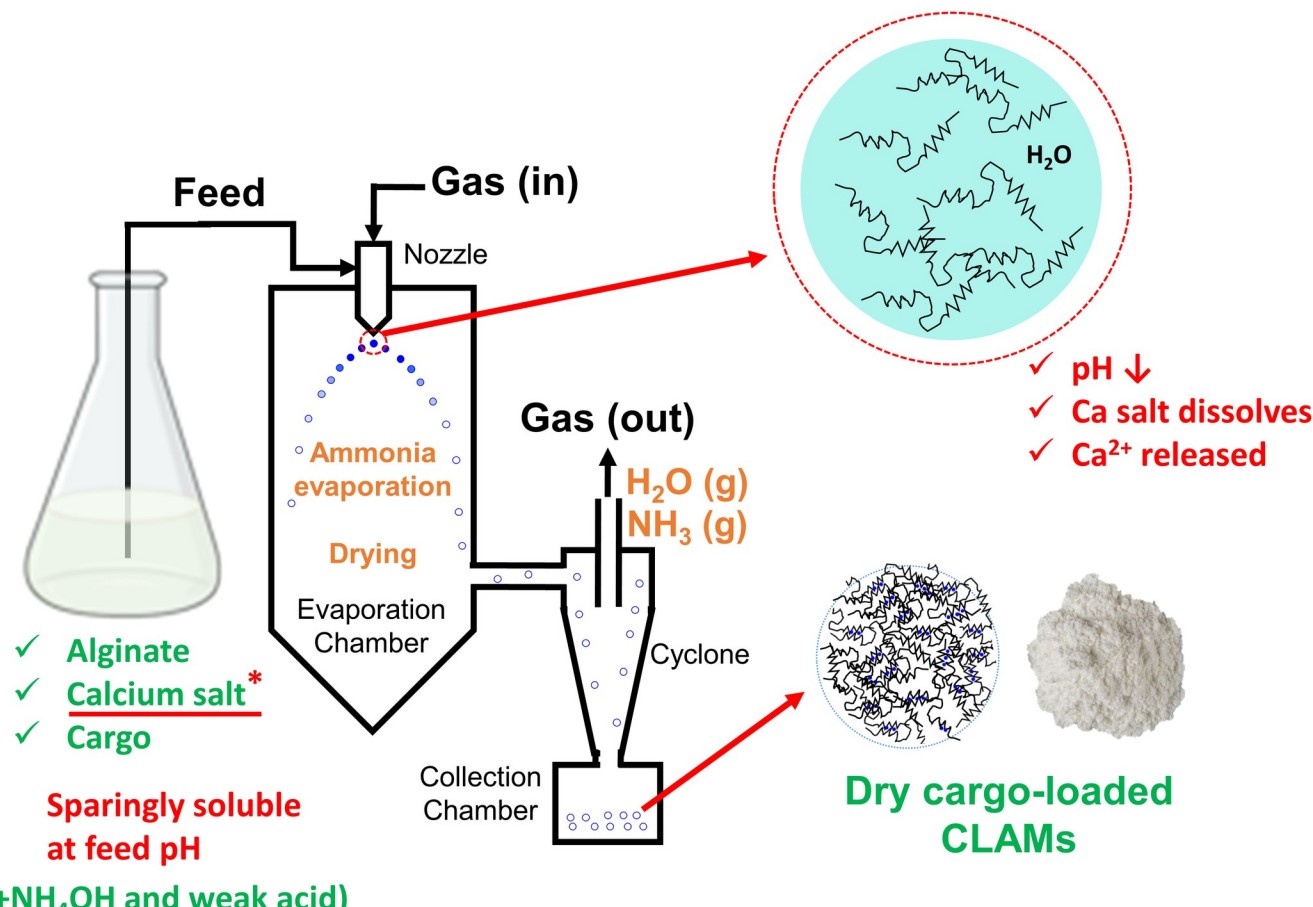

**Fig 1. Schematic of the spray-dried CLAMs process.** Alginate formulation containing insoluble calcium and a weak acid titrated with a volatile base is atomized in a spray dryer to produce crosslinked alginate microcapsules (CLAMs) in one step. The solution pH drops upon volatilization of the base after atomization, solubilizing calcium to interact with and crosslink negatively charged alginates.

previously described [10]. The extent of crosslinking is the percentage of alginates that remain insoluble when CLAMs are suspended in water under agitation for 2 h. The insoluble fraction was calculated from measurements of the solubilized alginate fraction relative to the total (soluble and insoluble) alginate present. Alginate concentrations were determined by periodic acid Schiff assay as described previously [10]. CLAMs created with varying alginate sources were measured against standard curves of their respective alginate types.

*Attenuated total reflectance (ATR)–FTIR characterization of alginates and CLAMs.* ATR-FTIR spectra of sodium alginate, calcium phosphate, succinic acid and CLAMs were collected using a single bounce total reflectance attenuation (ATR) cell equipped with a diamond internal reflection element and DMCTA detector (PIKE Technologies GladiATR). The spectra were recorded on a Thermo Nicolet 6700 FTIR spectrometer (Thermo Scientific) in the absorption mode in the range of 600–4000 cm$^{-1}$, with 128 scans at a resolution of 4 cm$^{-1}$.

## Statistical analysis

Statistical analyses of the data were conducted using the Data Analysis Toolpak in Excel. The null hypothesis of no difference in sample means were tested by the Student's t-Test assuming unpaired sets with unequal variances.

## Results and discussion

### The 'extent of cross-linking' in CLAMs formed by *in situ* internal gelation during spray drying

Crosslinked alginate microencapsulation by in situ internal gelation during spray drying (the 'CLAMs process') relies on pH-responsive solubility of calcium salts to allow ion-mediated gelling of alginates only under lower pH conditions after volatilization of the base upon spray atomization of the feed (Fig 1) [20]. One measure of the 'extent of crosslinking' achieved in the crosslinked alginate microcapsules (CLAMs) is the insoluble fraction of the powder in water [7,9]. For example, when a CLAMs sample is suspended in water and equilibrated under agitation, dissolution of 20% of alginates from the powder sample is taken as an indication that the remaining 80% of the alginates were insoluble because of calcium crosslinking. Thus, in this example, the extent of crosslinking of the CLAMs sample is 80%. The extent of crosslinking assessed by this insolubility metric can be controlled by varying the calcium content (i.e. the calcium to alginate ratio) in the spray drying feed formulation [10]. Within a finite range, higher calcium loadings lead to greater extents of crosslinking [10] (**Fig 2**).

The range and the maximum extents of crosslinking achievable by varying calcium loading in the feed, however, depends on the source of alginates used. For example, four commercially-sourced alginates (Table 1) resulted in different crosslinking extents under similar preparation conditions and calcium loading ranges (**Fig 2**). The alginate obtained from Sigma (LV, Table 1) exhibited a narrow range of 58–75% crosslinking for calcium/alginate ratios of 0.05–0.25. In contrast, for the same calcium/alginate ratios, the alginate from DuPont Danisco (HV3, Table 1) resulted in CLAMs with crosslinking extents of 33 to 100%. The BASF Hydagen alginate (HV1, Table 1) trended similarly to HV3 with a broad range of crosslinking from 33–94%. The TIC Algin 400 alginate (HV2, Table 1) resulted in a crosslinking range of 58–96%. Interestingly, for all four alginates, the extents of crosslinking plateaued at 0.125 calcium/alginate ratios, suggesting a saturation calcium loading within the parameters of this study. This is also the case for the LV alginate despite only reaching 75% crosslinking.

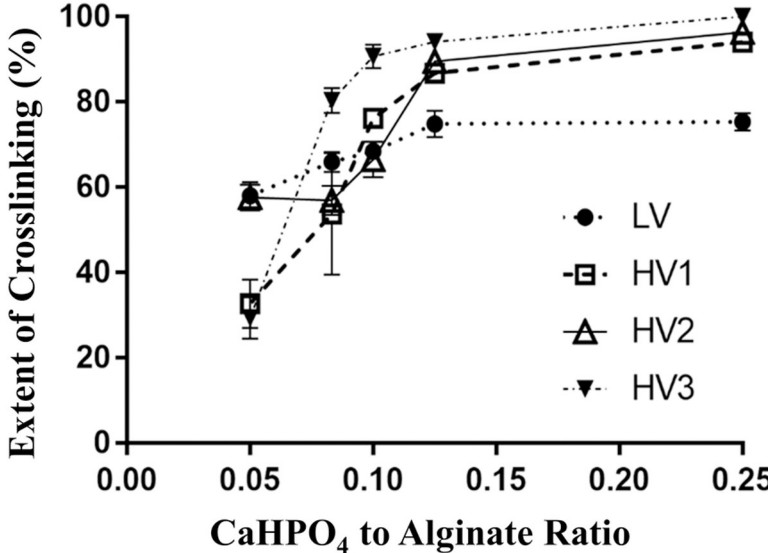

**Fig 2. The extent of cross-linking measured as the % of insoluble alginates in CLAMs samples as a function of calcium/alginate ratios in the spray dryer feed.** Error bars represent standard deviations of four replicates.

What influences an alginate's potential to form calcium-mediated crosslinks? Average polymer size, composition, and viscosity have all been demonstrated to impact the ability of alginates to gel in the presence of multivalent ions [1,30]. The general lack of available information on the properties of commercial alginates, however, poses a challenge in achieving predictable extents of crosslinking in CLAMs.

## Commercial alginate properties influencing crosslinking potential

The commercial alginates tested in these studies were all isolated from the cell walls of marine macroalgae, but are marketed for various end-uses (Table 1). The algae source can influence the composition of alginates such as the ratio of guluronates to mannuronates (G/M ratio) [25,31]; purification processing conditions can influence polymer size and size distributions [27,32]. We measured these properties to assess their role in influencing crosslinking in the CLAMs (Table 2).

The (weight averaged) molecular weights of the alginates ranged from 51–173 kDa, within previously reported ranges for commercial alginates [33] (Table 2). The LV alginates (51 kDa) were three-folds smaller than the HV1, HV2 or HV3 alginates (149, 164, and 173 kDa, respectively). Correspondingly, the hydrodynamic radius of LV alginates of 11 nm was nearly 3-folds smaller than the ~ 30 nm of HV1, HV2 and HV3 alginates. CLAMs formed with LV1 alginates exhibited a maximum crosslinking extent of 75%, while HV1, HV2 and HV3 CLAMs achieved ≥ 94% crosslinking (**Fig 2**).

The maximum potential crosslinking extent appear to correlate positively with alginate molecular weights above saturation calcium content in the feed (Fig 3A). While it also appears that narrower size range favors higher crosslinking extents (Fig 3B), decreasing dispersities trended strongly with increasing molecular weights ($R^2 = 0.94$, plot not shown), which precludes independent conclusions about the role of alginate size distribution on crosslinking from this sample set. The correlation between molecular weight and crosslinking only manifested at or above saturating calcium to alginate ratios. A possible explanation for the little to no correlation between molecular weights and crosslinking below saturation ratios of calcium

**Table 2. Properties of alginates used in this study (commercial source of alginates provided in Table 1).**

| Sample ID | $\overline{M}_w$ [1] (kDa) | Dispersity $\overline{M}_w/\overline{M}_n$ [2] | $R_h$ [3] (nm) | Intrinsic viscosity [4] (dL/g) | Apparent viscosity at feed concentration (mPa*s)[5] | pH [6] | G/M ratio[7] |
|---|---|---|---|---|---|---|---|
| **LV** | 51[a] (0.3) | 1.9[a] (0.003) | 11[a] (0.03) | 2.3[a] (0.009) | 74 (0.4) | 5.5[a] (0.01) | 3[a] (0.09) |
| **HV1** | 149[b] (0.09) | 1.6[b] (0.02) | 27[b] (0.2) | 9.3[b] (0.1) | 26 (0.2) | 6.4[b] (0.05) | 3[a](0.1) |
| **HV2** | 164[c] (0.4) | 1.4[c] (0.02) | 30[c] (0.01) | 11.1[c] (0.05) | 45 (0.3) | 7.0[c] (0.03) | 3[a] (0.04) |
| **HV3** | 173[d] (0.5) | 1.4[c,d] (0.02) | 29[d] (0.1) | 10.6[d] (0.05) | 37 (0.3) | 6.3[c,d] (0.03) | 3[a] (0.05) |

[1] Weight averaged molecular weight, $\overline{M}_w$, determined by SEC with right angle light scattering.

[2] Dispersity determined as a ratio of $\overline{M}_w$ to the number averaged molecular weight ($\overline{M}_n$).

[3] Hydrodynamic radius, $R_h$, is obtained by integrating under the curve calculated from the molecular weight (M) and intrinsic viscosity by Einstein's viscosity equation:
$[\eta]M = \frac{\frac{10}{3}\pi R_h^3}{N_A}$ at each data slice (n = 3).

[4] Intrinsic viscosity was measured by inline solution viscometry.

[5] LV at 2% (w/w), and HV1, HV2 and HV3 at 0.5% (w/w) in water, measured by falling ball viscometry (n = 6). Apparent viscosities at varying solution concentrations given in S1 File.

[6] 1% solution in nanopure water (n = 3).

[7] From HPLC analysis of M residues and total monomers of acid hydrolysates (see S1 File) (n = 8).

Standard deviations in parentheses (number of replicates (n) for each measurement are given in the footnotes). Similar superscript letters within each column signify no statistically significant differences between sample means with a p<0.05 by Student's t-test.

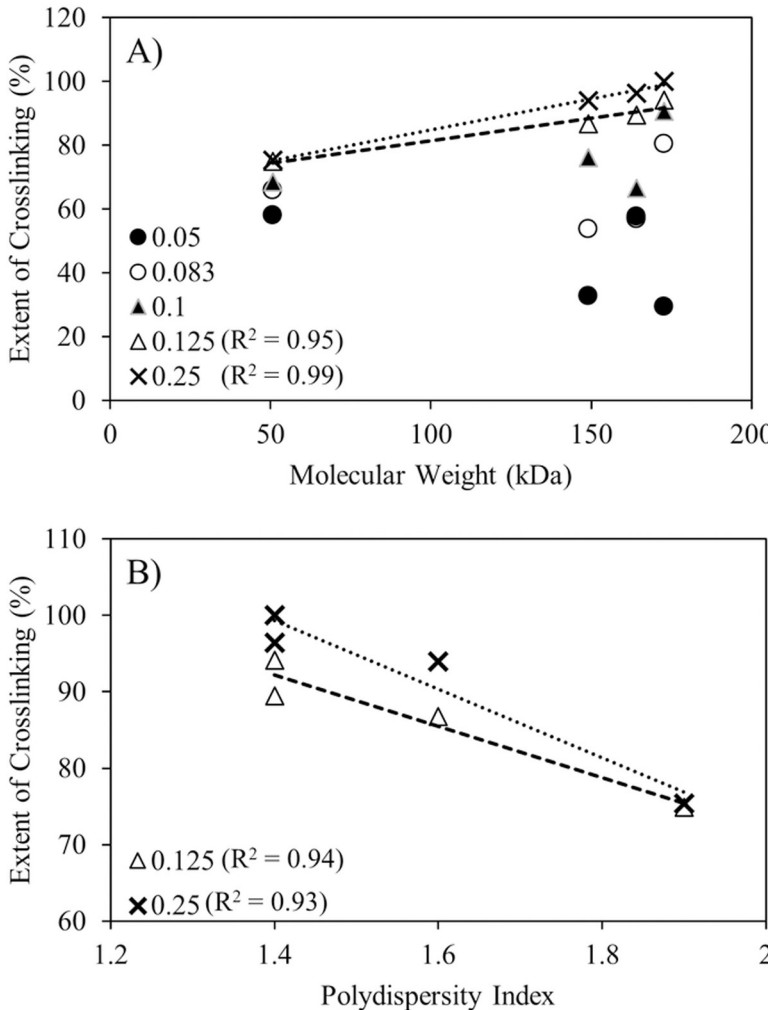

**Fig 3.** The extent of crosslinking achieved in CLAMs at varying calcium to alginate ratios with respect to A) molecular weight and B) polydispersity of the alginates. Calcium to alginate ratios given in the legends. The $R^2$ values for linear regressions shown in the graphs are next to the corresponding legend.

in the formulation (Fig 3A) is because of fewer and more sparsely distributed $CaHPO_4$ salt crystals in each atomized droplet at lower concentrations in the feed. The $CaHPO_4$ used in this study has a broad size distribution ranging from ~ 0.1 - ~60 μm [10]. Once the feed is atomized in the spray dryer, calcium dissolution and diffusion must occur within the few seconds residence time in the evaporation chamber of the spray dryer, where rapid moisture removal is occurring simultaneously. At low loadings, calcium crosslinking most likely only occurs between alginates in the immediate vicinity of the salt crystals. Thus in the CLAMs process, alginate molecular weight may only limit crosslinking at higher calcium loadings not dominated by dissolution, diffusion and drying kinetics.

## The role of alginate composition on crosslinking in CLAMs

Polyguluronate (PolyG) segments of alginates preferentially electrostatically crosslink via calcium or other divalent cations in the "egg-box" conformation [34]. Conversely, the more extended and 'flatter' chain configuration of polymannuronate (PolyM) segments is thought

to disfavor calcium crosslinking [35]. While crosslinks between PolyG segments contribute to gel stiffness, PolyM segments along with mixed M G segments have been shown to moderate elasticity and contribute to overall chain associations [23]. The four commercial alginates had similar average ratios of G to M residues (G/M ratio ~ 3) (Table 2), though the molecular arrangement was not determined.

To study the contribution of G/M ratios and molecular arrangements to crosslinking, CLAMs were formed using varying ratios of PolyM and PolyG alginates [21]. The expectation was that CLAMs with lower fractions of PolyG would exhibit less crosslinking. Two methods of CLAMs formation were tested–the conventional external gelation method where solutions of alginates were dripped into a calcium chloride bath then dried, and the CLAMs process. Indeed, by external gelation, only 5 ± 1% crosslinking was achieved with PolyMs while 93 ± 1% was seen for PolyGs (**Fig 4A**). Dripping PolyM alginates into a calcium bath resulted in minimal gelation, while dripping PolyG alginates rapidly formed stiff gels. When the CLAMs were formed by spray drying however, no significant differences in crosslinking between PolyM and PolyG CLAMs were found (50 ± 12% and 65 ± 13% for PolyM and PolyG CLAMs, respectively) (**Fig 4A**). Contrary to expectations, PolyM CLAMs formed by spray drying were significantly gelled and insoluble in water.

In fact, when comparing crosslinking extent for CLAMs made entirely of PolyM, or entirely of PolyG, or varying ratios of PolyG to PolyM, no statistical difference was found (**Fig 4B**). This is a surprising result, because if the microcapsule wall was composed mostly of calcium ion-mediated linkages, the PolyM chains should show essentially no ability to gel. Gelation in the PolyM particles formed by spray-drying thus suggests additional gelation mechanisms facilitated by the CLAMs process. In addition to ion-mediated crosslinking, alginates can also form insoluble acid gels stabilized by intermolecular hydrogen bonds. The conventional external gelation method does not facilitate alginic acid gelation unless the pH of the calcium chloride bath is lower than the $pK_a$ of the alginates. The CLAMs process, however, could facilitate localized alginic acid gelation within the particles if the simultaneous drying and pH drop during spray drying causes localized 'hot spots' of highly concentrated protons.

## Simultaneous *in situ* gelation and drying facilitates calcium crosslinking and acid gel formation

The FTIR spectra of the alginates and the spray dried CLAMs offered key insights into alginate interactions in CLAMs (Fig 5). In the 1200–1800 cm$^{-1}$ wavenumber region of the FTIR spectra,

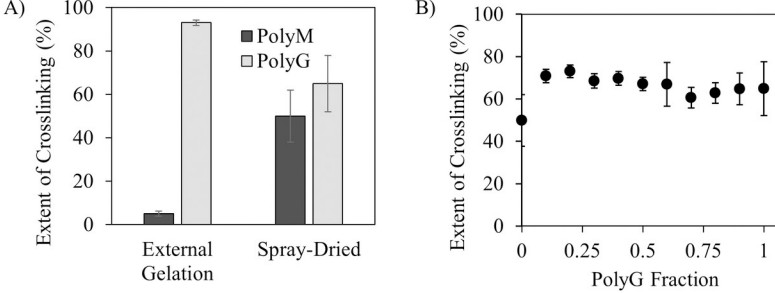

**Fig 4. Crosslinking as a function of polyguluronate (PolyG) fraction.** A) Crosslinking of external gelation formed and spray-dried powders with 100% PolyM or 100% PolyG (n = 3); B) Crosslinking of spray dried CLAMs formed with ratios ranging from PolyG fraction of 0 (0:1 PolyG to PolyM) to 1 (1: 0 PolyG to PolyM). The difference in crosslinking between PolyG = 0 (100% PolyM) and PolyG = 1 (100% PolyG) cross-linking was not found to be statistically significant (p-value = 0.0563).

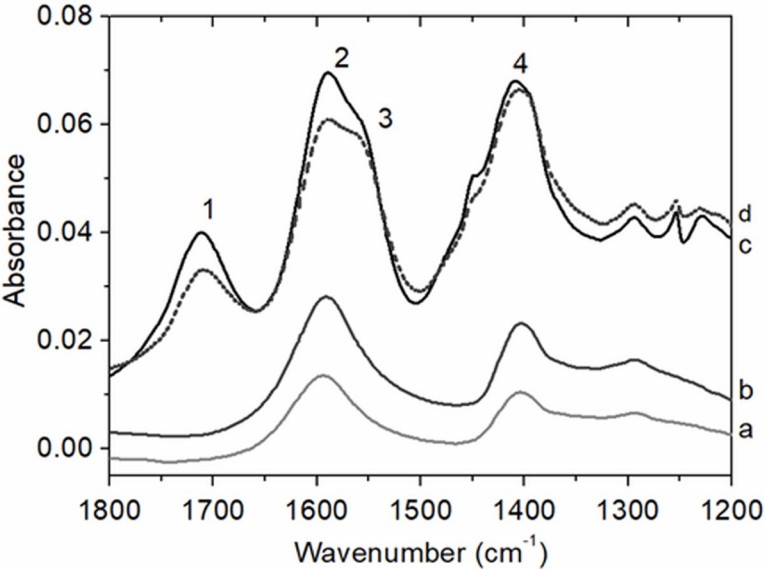

**Fig 5. ATR-FTIR spectra of alginates (a: LV and b: HV3) and CLAMs (c: HV3 CLAMs and d: LV CLAMs) in the 1200–1800 cm$^{-1}$ range.** Both alginates and CLAMs spectra contain absorption peaks for asymmetric and symmetric COO- stretching vibration (centered at 1595 cm$^{-1}$ (peak 2) and 1410 cm$^{-1}$ (peak 4), respectively) indicating the presence of alginate salts. The appearance of an absorption peak for carboxylic acid at 1710 cm$^{-1}$ (peak 1) in the CLAMs spectra is due to the presence of alginic acids. Additionally the peak centered at 1550 cm$^{-1}$ (peak 3) in the CLAMs spectra indicate the formation of an amine salt.

LV and HV3 alginates, as well as the CLAMs formed with these alginates absorbed strongly at 1595 cm$^{-1}$ (peak 2) and 1410 cm$^{-1}$ (peak 4) due to the symmetric and asymmetric stretching of the ionized carboxylate. The presence of COO$^-$ groups are expected in the sodium alginates (-COO$^-$ Na$^+$) [36], and in electrostatically crosslinked CLAMs (-COO$^-$ Ca$^{++}$). In the CLAMs spectra (but absent in the alginate spectra) is an additional strongly absorbing peak centered at 1710 cm$^{-1}$ (peak 1) indicating the presence of carboxylic acid. Under high acid concentrations and as the pH of the surrounding solution approaches the pK$_a$ of alginates, protonation of the carboxyl groups occurs. The FTIR spectra of CLAMs indicate that a fraction of the carboxyl groups become protonated during formation; i.e., alginic acid is formed. While the formulation also contained succinic acid, the absorption maximum of its carbonyl peak is at 1685 cm$^{-1}$ and offset from that of the CLAMs sample (S1 File).

Calcium-mediated electrostatic crosslinking in spray dried CLAMs occur when the pH decreases in each atomized droplet to solubilize the calcium salt. The pH drop is due to the volatilization of the base, and the pH in the droplet is buffered by the pK$_a$ of a weak acid. When the spray dried powder is dissolved in water, the supernatant pH poises near the pK$_a$ of the weak acid, confirming the role of the acid in the particles. The weak acid, in this case succinic acid (pK$_{a,2}$ = 4.2), was chosen to maintain the pH above the alginate pK$_a$ (~3.5) such that the polyuronates are negatively charged and available for electrostatic associations with divalent calcium ions. In the spray drying process, pH decrease and salt dissolution occur at the same time that the droplets are rapidly drying and shrinking (in a matter of seconds in the bench scale spray dryer). We speculate that the rapid removal of water from the droplets result in localized regions of highly concentrated protons; i.e. while the average pH remain near the pK$_a$ of the acid, there may be localized regions where pH $<<$ pK$_{a,alginates}$ simply due to water removal. Additionally, hydrostatic pressures by rapid particle shrinkage facilitate chain-chain associations that do not occur when alginate solutions are dripped or sprayed into calcium

baths. Taken together, we speculate that during spray dry CLAMs formation, in addition to calcium crosslinking, insoluble alginic acid gelation occurs.

## Linking alginate molecular property with bulk property influencing gelation in CLAMs

The water-insoluble fraction of CLAMs formed by in situ internal gelation during spray drying is due to both calcium crosslinked alginates and alginic acids. Thus, for spray dried CLAMs, the 'extent of gelation' is a more appropriate characterization of the insoluble fraction that accounts for both gelation mechanisms. At or above saturating calcium ratios, the extent of gelation of CLAMs can be controlled on the basis of the alginate molecular weights. However, commercial alginates rarely include molecular weight information; instead, bulk solution viscosities are more commonly provided. The molecular weight of polymers is typically closely correlated to the viscosity of the polymer solution [37]. The alginates in this study were no exception where both the intrinsic viscosity and the apparent viscosities trended with molecular weight (Fig 6). Thus, one can use viscosity as an alginate selection criterion to target higher or lower gelation extents in CLAMs.

## Conclusions

The purpose of this work was to understand limiting factors controlling crosslinking extents in CLAMs formed by in situ gelation during spray drying. While molecular weights, alginate size distributions and molecular compositions were explored, the surprise finding was the formation of alginic acids in CLAMs. In conventional external gelation methods, where alginate solutions are dripped into calcium salt solutions, alginic acids do not form at the typical neutral pHs used. In the case of spray-dried CLAMs, however, small aliquots of alginates, calcium salt and organic acids are concentrated rapidly, giving rise to the possibility of localized spikes in proton concentrations to protonate the alginates. Alginic acids, being insoluble in water, thus likely contribute to the measure of the extent of crosslinking in spray-dried CLAMs. Moreover, the combination of calcium crosslinking and alginic acid formation in CLAMs was not influenced by the molecular composition of alginates. It thus appears that while

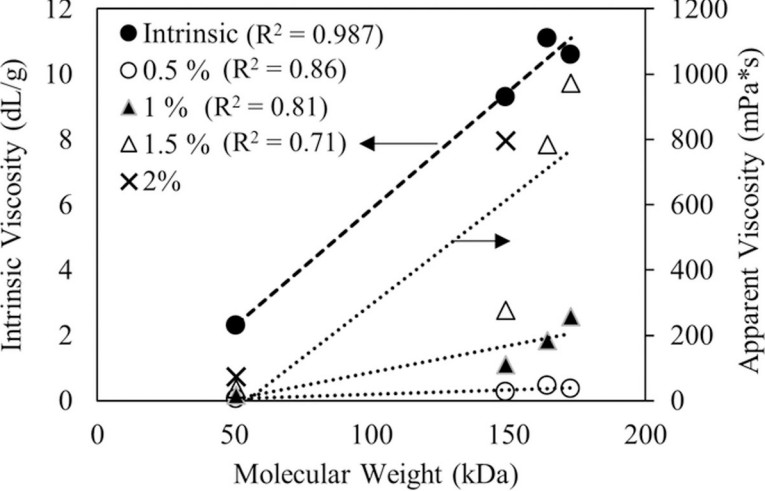

**Fig 6. Intrinsic viscosity of alginates (primary axis) and apparent viscosities of alginate solutions at varying concentrations (secondary axis) plotted with respect to average molecular weights.** $R^2$ values of regression lines are shown next to the corresponding legend. See S1 File for additional information on alginate solution viscosities.

conventional external gelation methods require careful consideration of the alginate composition to control the extents of crosslinking achieved, molecular composition is not an important property for the insolubility of spray-dried CLAMs. Rather, 'crosslinking' or more accurately the 'gelation' of spray-dried CLAMs, can best be controlled by careful selection of alginate molecular weights at saturating calcium salt concentrations. As molecular weights and viscosity of alginates are closely correlated, the selection criterion for commercial alginates to influence gelation in the CLAMs process is viscosity.

## Supporting information

**S1 File.**
(DOCX)

## Acknowledgments

The authors acknowledge Dr. Sanjai Parikh for the use of his ATR-FTIR, and Ted Diesenroth and Rupa Darji at BASF for valuable discussions. The authors thank Malvern Panalytical for providing GPC/SEC data for this study.

## Author Contributions

**Conceptualization:** Tina Jeoh, Dana E. Wong, Scott A. Strobel, Herbert B. Scher.

**Formal analysis:** Dana E. Wong, Kevin Hudnall, Nadia R. Pereira.

**Funding acquisition:** Tina Jeoh.

**Investigation:** Dana E. Wong, Kevin Hudnall, Nadia R. Pereira, Kyle A. Williams, Benjamin M. Arbaugh, Julia C. Cunniffe.

**Methodology:** Dana E. Wong, Scott A. Strobel, Kevin Hudnall.

**Project administration:** Tina Jeoh.

**Resources:** Tina Jeoh.

**Supervision:** Tina Jeoh.

**Visualization:** Tina Jeoh.

**Writing – original draft:** Dana E. Wong.

**Writing – review & editing:** Tina Jeoh, Dana E. Wong, Scott A. Strobel, Kevin Hudnall, Nadia R. Pereira, Kyle A. Williams, Benjamin M. Arbaugh, Julia C. Cunniffe, Herbert B. Scher.

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
