## [Decision Letter · Decision Letter 0]

14 Oct 2020

PONE-D-20-29628

How alginate properties influence in situ internal gelation in Crosslinked Alginate Microcapsules (CLAMs) formed by spray drying

PLOS ONE

Dear Dr. Jeoh,

Thank you for submitting your manuscript to PLOS ONE. After careful consideration, we feel that it has merit but does not fully meet PLOS ONE’s publication criteria as it currently stands. Therefore, we invite you to submit a revised version of the manuscript that addresses the points raised during the review process.

We look forward to receiving your revised manuscript.

Kind regards,

C. Anandharamakrishnan

Academic Editor

PLOS ONE

Journal Requirements:

2. Please amend the manuscript submission data (via Edit Submission) to correct the corresponding author name, this should be listed as "Tina Jeoh" and not "Tina Jeoh Jeoh".

'The authors received no specific funding for this work.

SAS was supported the Agriculture and Food Research Initiative grant 2018-67012-28029 from the USDA National Institute of Food and Agriculture

The funders had no role in study design, data collection and analysis, decision to publish, or preparation of the manuscript.'

We note that one or more of the authors are employed by a commercial company: Malvern Panalytical.

Within your Competing Interests Statement, please confirm that this commercial affiliation does not alter your adherence to all PLOS ONE policies on sharing data and materials by including the following statement: "This does not alter our adherence to  PLOS ONE policies on sharing data and materials.” (as detailed online in our guide for authors http://journals.plos.org/plosone/s/competing-interests) . If this adherence statement is not accurate and  there are restrictions on sharing of data and/or materials, please state these.

We also note that you have a patent relating to material pertinent to this article.

Please declare this patent (with details including name and number), in your amended statement of Competing Interests, along with any other relevant declarations relating to employment, consultancy, patents, products in development or modified products etc.

Please confirm that this does not alter your adherence to all PLOS ONE policies on sharing data and materials, as detailed online in our guide for authors http://journals.plos.org/plosone/s/competing-interests by including the following statement: "This does not alter our adherence to  PLOS ONE policies on sharing data and materials.”

If there are restrictions on sharing of data and/or materials, please state these.

Please note that we cannot proceed with consideration of your article until this information has been declared.

<h3>** **</h3>

<h3>**Please know it is PLOS ONE policy for corresponding authors to declare, on behalf of all authors, all potential competing interests for the purposes of transparency. PLOS defines a competing interest as anything that interferes with, or could reasonably be perceived as interfering with, the full and objective presentation, peer review, editorial decision-making, or publication of research or non-research articles submitted to one of the journals. Competing interests can be financial or non-financial, professional, or personal. Competing interests can arise in relationship to an organization or another person. Please follow this link to our website for more details on competing interests: http://journals.plos.org/plosone/s/competing-interests**</h3>

4. Please ensure that you refer to Figure 5 in your text as, if accepted, production will need this reference to link the reader to the figure.

5. Please include captions for your Supporting Information files at the end of your manuscript, and update any in-text citations to match accordingly. Please see our Supporting Information guidelines for more information: http://journals.plos.org/plosone/s/supporting-information

Reviewers' comments:

Reviewer's Responses to Questions

**Comments to the Author**

1. Is the manuscript technically sound, and do the data support the conclusions?

Reviewer #1: Yes

Reviewer #2: Partly

Reviewer #3: Yes

Reviewer #4: Yes

2. Has the statistical analysis been performed appropriately and rigorously? 

Reviewer #1: Yes

Reviewer #2: No

Reviewer #3: Yes

Reviewer #4: Yes

3. Have the authors made all data underlying the findings in their manuscript fully available?

Reviewer #1: Yes

Reviewer #2: No

Reviewer #3: Yes

Reviewer #4: No

4. Is the manuscript presented in an intelligible fashion and written in standard English?

Reviewer #1: Yes

Reviewer #2: Yes

Reviewer #3: Yes

Reviewer #4: Yes

5. Review Comments to the Author

Reviewer #1: Dear Editor,

Thank you for giving me opportunity for review this manuscript.

This manuscript focused on crosslinking of alginates. It is well written and results are explained well and supported by suitable literature. The research question is clearly mentioned. I found few points need to be add which improve the article. For instance

Line 91: What are others methods of crosslinking other than spray drying, kindly mention that

Line 150: Kindly mention reference for this method if authors have developed the method then kindly mention that

Line 187: What is basis for selection of inlet temperature? Does inlet temperature affect the alginate crosslinking?

Line 188: What is outlet temperature? Does outlet temperature affect the alginate crosslinking?

Line 197: Kindly give reference for the method

Line 225: What is yield during spray if any losses are there kindly mention in manuscript

Line 236: Authors have taken calcium phosphate for this study, why calcium phosphate has been considered? Does salt type influence the crosslinking?

Line 249: There is any method to get 100 % crosslinking

Line 256: What are other properties of alginate which may influence the crosslinking

Line 367: What is moisture content of CLAMs does moisture content of CLAMs affect the crosslinking

Line 378: kindly mention the limitations of this study if any

Reviewer #2: The authors studied about “How alginate properties influence in situ internal gelation in Crosslinked Alginate Microcapsules (CLAMs) formed by spray drying”. This study is performed alternative to external gelation to prove the advantage of this study. This represented study has to clarify the following points before acceptance in this journal:

1. The microcapsule preparation method used calcium as crosslinker, the authors should aware the used calcium concentration should not exceed physiological intake limit. Excessive calcium consumption will cause severe health complication specially cardiac complications. Hence, explain in the manuscript how much quantity of calcium used and that quantity are in acceptable limit or not.

2. Ammonium hydroxide is used as a base in this study to maintain higher pH. It is boiling point almost near to room temperature, so it can evaporate in room atmosphere before spray or it can completely evaporate in nozzle head. In this concern how ammonium hydroxide based pH theory works in microcapsule preparation. Please explain.

3. In page no:11, line no: 226 mentioned that crosslinked alginate microencapsulation. What type of molecule is encapsulated into the microcapsule?, like that in graphical abstract mentioned dry cargo-loaded CLAM. What type of cargo is loaded into the CLAM?

4. Extend of crosslinking data of external crosslinked microparticles is not mentioned in the manuscript and not compared with CLAM. Please include that in the manuscript.

Reviewer #3: The paper entitled “Alginate properties impacting gelation in spray-dried crosslinked alginate microcapsules” is related to influence of alginate molecular properties and their effect on extend of cross linking in in-situ gelation in CLAMs produced by spray drying technique.

The approach of the manuscript is interesting and could provide information. The paper is well written and methodology is understandable.

I would like to suggest the Editor to accept the manuscript with minor corrections.

1. Materials and methods require details in explaining the statistical analysis.

2. In table 2, authors explain significant difference only for G/M ratio. But they didn’t say anything about significant difference for other parameters in the table. Are there? Also please use same number of decimals for all values.

3. Figure 5 caption explains the same as in discussion part. Please change it.

Reviewer #4: Review Comments

Manuscript Title: How alginate properties influence in situ internal gelation in Crosslinked Alginate Microcapsules (CLAMs) formed by spray drying

The following aspects may be considered to improve the work.

1. Improve quality of all figures. Some values are just not readable!

2. Elaborate the underlying chemical changes as equations.

3. Improve discussion with recent (2019-20) articles on spray drying + crosslinking (perhaps even other molecules).

4. Add details on the purity and composition of all raw materials. How would they impact the findings of this study?

5. Add details on the drying mechanism in the spray dryer (relevant to the microcapsule formation).

6. What about the post-encapsulation quality?

6. PLOS authors have the option to publish the peer review history of their article (what does this mean?). If published, this will include your full peer review and any attached files.

Reviewer #1: **Yes: **PINTU CHOUDHARY

Reviewer #2: No

Reviewer #3: No

Reviewer #4: No

---

## [Author Response · Author response to Decision Letter 0]

23 Dec 2020

A document containing point-by-point response to reviewers has been uploaded.

---

## [Decision Letter · Decision Letter 1]

13 Jan 2021

PONE-D-20-29628R1

How alginate properties influence in situ internal gelation in Crosslinked Alginate Microcapsules (CLAMs) formed by spray drying

PLOS ONE

Dear Dr. Jeoh,

Thank you for submitting your manuscript to PLOS ONE. After careful consideration, we feel that it has merit but does not fully meet PLOS ONE’s publication criteria as it currently stands. Therefore, we invite you to submit a revised version of the manuscript that addresses the points raised during the review process.

We look forward to receiving your revised manuscript.

Kind regards,

C. Anandharamakrishnan

Academic Editor

PLOS ONE

Reviewers' comments:

Reviewer's Responses to Questions

**Comments to the Author**

1. If the authors have adequately addressed your comments raised in a previous round of review and you feel that this manuscript is now acceptable for publication, you may indicate that here to bypass the “Comments to the Author” section, enter your conflict of interest statement in the “Confidential to Editor” section, and submit your "Accept" recommendation.

Reviewer #2: All comments have been addressed

Reviewer #5: All comments have been addressed

Reviewer #6: All comments have been addressed

Reviewer #7: All comments have been addressed

Reviewer #8: All comments have been addressed

Reviewer #9: All comments have been addressed

Reviewer #10: (No Response)

Reviewer #11: (No Response)

2. Is the manuscript technically sound, and do the data support the conclusions?

Reviewer #2: Partly

Reviewer #5: Yes

Reviewer #6: Yes

Reviewer #7: Yes

Reviewer #8: Yes

Reviewer #9: Yes

Reviewer #10: Partly

Reviewer #11: Partly

3. Has the statistical analysis been performed appropriately and rigorously? 

Reviewer #2: Yes

Reviewer #5: Yes

Reviewer #6: Yes

Reviewer #7: Yes

Reviewer #8: Yes

Reviewer #9: Yes

Reviewer #10: N/A

Reviewer #11: N/A

4. Have the authors made all data underlying the findings in their manuscript fully available?

Reviewer #2: Yes

Reviewer #5: Yes

Reviewer #6: Yes

Reviewer #7: Yes

Reviewer #8: Yes

Reviewer #9: Yes

Reviewer #10: Yes

Reviewer #11: Yes

5. Is the manuscript presented in an intelligible fashion and written in standard English?

Reviewer #2: Yes

Reviewer #5: Yes

Reviewer #6: Yes

Reviewer #7: Yes

Reviewer #8: Yes

Reviewer #9: Yes

Reviewer #10: Yes

Reviewer #11: Yes

6. Review Comments to the Author

Reviewer #2: (No Response)

Reviewer #5: (No Response)

Reviewer #6: The authors did address all the comments by the reviewers sufficiently. I agree with the author that the comment by reviewer 3 regarding the caption of Figure 5 is not applicable since the Figure is very complex and an extensive caption to explain is necessary.

Reviewer #7: This manuscript reported that “How alginate properties influence in situ internal gelation in crosslinked alginate microcapsules(CLAMs) formed by spray drying”. The authors compared the effects of alginic acid molecular weight, size distribution and molecular composition on CLAMs gelation. Overall, the manuscript was well written, experiments were well designed, the data interpretation is sound and the conclusion drawn by authors is firmly supported by the data provided. Therefore, it would be suitable for publication in the journal "PLOS ONE" without further revision.

Reviewer #8: This work studies the impact of alginate properties in crosslinked alginate microcapsules formed by CLAMs process.

The authors described various factors that affect the extent of crosslinking, with a particular focus on the influence of commercially available alginates. Due to lack of information of commercial alginates, the authors performed a characterisation of these to study the impact of their properties on the extent of crosslinking in CLAMs.

Interestingly, the impact of alginate properties is dependent on the method utilised for crosslinking, where spray-dried CLAMs are mainly influenced by the molecular weight or density of alginates at saturating salt concentrations. The paper is well written and the materials and methods are clear and provide sufficient information.

This manuscript has been already reviewed and most of my doubts have been solved during this round of revision. I have only one question regarding the G/M ratio. The authors described that by external gelation, 5% and 93% of crosslinking is achieved with PolyMs and PolyGs, respectively. On the other hand, 50% crosslinking was observed with PolyMs and PolyGs by CLAMs process, which contradicts the expectations of the authors. I believe these results need to be further discussed by the authors. Is this equal crosslinking of polyMs and polyGs influenced by alginic acid? are there any other parameters that may influence the outcomes observed?.

Reviewer #9: The authors have adequately responded to all reviewer comments. The paper is well-written, and will be an important contribution to this area of pharmaceutical development, particle engineering and spray drying.

Reviewer #10: The authors present preparation of crosslinked alginate microcapsules (CLAMs) with alginates of different properties using spray drying technique. The study seems to have been designed well and the experiments are done rigorously. The authors seem to have addressed most of earlier concerns raised be reviewers. However, there are few concerns that need to be fixed before it is finalized.

Specific comments:

In the process of preparation of the CLAMs the feed suspension contains calcium hydrogen phosphate, succinic acid, and ammonium hydroxide along with Alginates. When the crosslinking takes place by calcium ions due to pH change, the crosslinking possibilities are; it can crosslink the alginates directly (alginate-Ca-alginate), or it can crosslink alginate through a succinate-bridge (alginate-Ca-succinate-Ca-alginate), or it can conjugate an alginate to one succinate carboxyl group and the other remains as carboxylic acid, or it can be a mixture of these. The COO- groups in succinates can exist in ionic form of COO- NH4+ (in presence of NH3 and H2O or NH4OH), and COO- H3O+ (on evaporation of NH3), and in equilibrium with COOH.

The mono carboxylic acid of the succinate can also be detectable at ~1710 cm-1 in the FTIR along with alginic acid if present. To confirm if the acid carbonyl observed is due to alginic or succinic or both, the process should be tested with and an alkyl acid containing only one carboxylic acid instead of a dicarboxylic acid in the process. Otherwise, the formation of alginic acid should be confirmed with another method of analysis to eliminate other possibilities.

Reviewer #11: The authors present preparation of crosslinked alginate microcapsules (CLAMs) with alginates of different properties using spray drying technique. The study seems to have been designed well and the experiments are done rigorously. The authors seem to have addressed most of earlier concerns raised be reviewers. However, there are few concerns that need to be fixed before it is finalized.

Specific comments:

In the process of preparation of the CLAMs the feed suspension contains calcium hydrogen phosphate, succinic acid, and ammonium hydroxide along with Alginates. When the crosslinking takes place by calcium ions due to pH change, the crosslinking possibilities are; it can crosslink the alginates directly (alginate-Ca-alginate), or it can crosslink alginate through a succinate-bridge (alginate-Ca-succinate-Ca-alginate), or it can conjugate an alginate to one succinate carboxyl group and the other remains as carboxylic acid, or it can be a mixture of these. The COO- groups in succinates can exist in ionic form of COO- NH4+ (in presence of NH3 and H2O or NH4OH), and COO- H3O+ (on evaporation of NH3), and in equilibrium with COOH.

The mono carboxylic acid of the succinate can also be detectable at ~1710 cm-1 in the FTIR along with alginic acid if present. To confirm if the acid carbonyl observed is due to alginic or succinic or both, the process should be tested with and an alkyl acid containing only one carboxylic acid instead of a dicarboxylic acid in the process. Otherwise, the formation of alginic acid should be confirmed with another method of analysis to eliminate other possibilities.

7. PLOS authors have the option to publish the peer review history of their article (what does this mean?). If published, this will include your full peer review and any attached files.

Reviewer #2: No

Reviewer #5: **Yes: **elham khodaverdi

Reviewer #6: **Yes: **Janke Kleynhans

Reviewer #7: No

Reviewer #8: **Yes: **Paulina Ramírez-García

Reviewer #9: No

Reviewer #10: No

Reviewer #11: No

---

## [Author Response · Author response to Decision Letter 1]

31 Jan 2021

Point-by-point response to reviewer comments are all included in the attached document. Thank you to the reviewers for their time and attention!

---

## [Editor Report · Decision Letter 2]

3 Feb 2021

How alginate properties influence in situ internal gelation in Crosslinked Alginate Microcapsules (CLAMs) formed by spray drying

PONE-D-20-29628R2

Dear Dr. Jeoh,

We’re pleased to inform you that your manuscript has been judged scientifically suitable for publication and will be formally accepted for publication once it meets all outstanding technical requirements.

Kind regards,

C. Anandharamakrishnan

Academic Editor

PLOS ONE

---

## [Editor Report · Acceptance letter]

11 Feb 2021

PONE-D-20-29628R2 

How alginate properties influence in situ internal gelation in crosslinked alginate microcapsules (CLAMs) formed by spray drying 

Dear Dr. Jeoh:

I'm pleased to inform you that your manuscript has been deemed suitable for publication in PLOS ONE. Congratulations! Your manuscript is now with our production department. 

Kind regards, 

on behalf of

Dr. C. Anandharamakrishnan 

Academic Editor

PLOS ONE